# A High-Quality *De novo* Genome Assembly from a Single Mosquito Using PacBio Sequencing

**DOI:** 10.3390/genes10010062

**Published:** 2019-01-18

**Authors:** Sarah B. Kingan, Haynes Heaton, Juliana Cudini, Christine C. Lambert, Primo Baybayan, Brendan D. Galvin, Richard Durbin, Jonas Korlach, Mara K. N. Lawniczak

**Affiliations:** 1Pacific Biosciences, 1305 O’Brien Drive, Menlo Park, CA 94025, USA; skingan@pacb.com (S.B.K.); clambert@pacb.com (C.C.L.); pbaybayan@pacb.com (P.B.); bgalvin@pacb.com (B.D.G.); 2Wellcome Sanger Institute, Wellcome Genome Campus, Hinxton CB10 1SA, UK; whh28@cam.ac.uk (H.H.); jc39@sanger.ac.uk (J.C.); 3Department of Genetics, University of Cambridge, Downing Street, Cambridge CB2 3EH, UK; rd109@cam.ac.uk

**Keywords:** low-input DNA, *de novo* genome assembly, long-read SMRT sequencing, mosquito

## Abstract

A high-quality reference genome is a fundamental resource for functional genetics, comparative genomics, and population genomics, and is increasingly important for conservation biology. PacBio Single Molecule, Real-Time (SMRT) sequencing generates long reads with uniform coverage and high consensus accuracy, making it a powerful technology for *de novo* genome assembly. Improvements in throughput and concomitant reductions in cost have made PacBio an attractive core technology for many large genome initiatives, however, relatively high DNA input requirements (~5 µg for standard library protocol) have placed PacBio out of reach for many projects on small organisms that have lower DNA content, or on projects with limited input DNA for other reasons. Here we present a high-quality *de novo* genome assembly from a single *Anopheles coluzzii* mosquito. A modified SMRTbell library construction protocol without DNA shearing and size selection was used to generate a SMRTbell library from just 100 ng of starting genomic DNA. The sample was run on the Sequel System with chemistry 3.0 and software v6.0, generating, on average, 25 Gb of sequence per SMRT Cell with 20 h movies, followed by diploid *de novo* genome assembly with FALCON-Unzip. The resulting curated assembly had high contiguity (contig N50 3.5 Mb) and completeness (more than 98% of conserved genes were present and full-length). In addition, this single-insect assembly now places 667 (>90%) of formerly unplaced genes into their appropriate chromosomal contexts in the AgamP4 PEST reference. We were also able to resolve maternal and paternal haplotypes for over 1/3 of the genome. By sequencing and assembling material from a single diploid individual, only two haplotypes were present, simplifying the assembly process compared to samples from multiple pooled individuals. The method presented here can be applied to samples with starting DNA amounts as low as 100 ng per 1 Gb genome size. This new low-input approach puts PacBio-based assemblies in reach for small highly heterozygous organisms that comprise much of the diversity of life.

## 1. Introduction

Exciting efforts to sequence the diversity of life are building momentum [1] but one of many challenges that these efforts face is the small size of most organisms. For example, arthropods, which comprise the most diverse animal phylum, are typically small. Beyond this, while levels of heterozygosity within species vary widely across taxa, intraspecific genetic variation is often highest in small organisms [2]. Over the past two decades, reference genomes for many small organisms have been built through considerable efforts of inbreeding organisms to reduce their heterozygosity levels such that many individuals can be pooled together for DNA extractions. This approach has varied in its success, for example working well for organisms that are easy to inbreed (e.g., many *Drosophila* species [3]), but less well for species that are difficult or impossible to inbreed (e.g., *Anopheles* [4]). Therefore, many efforts to sequence genomes of small organisms have relied primarily on short-read approaches due to the large amounts of DNA required for long-read approaches. For example, the recent release of 28 arthropod genomes as part of the i5K initiative used four different insert size Illumina libraries, resulting in an average contig N50 of 15 kb and scaffold N50 of 1 Mb [5].

Another way to overcome DNA input requirements, while also reducing the number of haplotypes present in a DNA pool, is to limit the number of haplotypes in the pool of individuals by using offspring from a single cross. This is easier than multiple generations of inbreeding, and can be successful. For example, a recent PacBio *Aedes aegypti* assembly used DNA extracted from the offspring of a single cross, thus reducing the maximum number of haplotypes for any given locus to four, thereby improving the assembly process and achieving a contig N50 of 1.3 Mb [6].

However, for an initiative like the Earth BioGenome Project [1] that aims to build high-quality reference genomes for more than a million described species over the next decade, generating broods to reach sufficient levels of high molecular weight DNA for long-read sequencing will be infeasible for the vast majority of organisms. Therefore, new methods that overcome the need to pool organisms are needed to support the creation of reference-quality genomes from wild-caught individuals to increase the diversity of life for which reference genomes can be assembled. Here, we present the first high-quality genome assembled with unamplified DNA from a single individual insect using a new workflow that greatly reduces input DNA requirements.

## 2. Materials and Methods

### 2.1. DNA Isolation and Evaluation

High molecular weight DNA was isolated from a single *Anopheles coluzzii* female from the Ngousso colony. This colony was created in 2006 from the broods of approximately 100 wild-caught pure *An. coluzzii* females in Cameroon (pers. comm. Anna Cohuet). Although the colony has been typically held at >100 breeding individuals, given the long time since colonization, there is undoubtedly inbreeding. A single female was ground in 200 µl PBS using a pestle with several up and down strokes (i.e., no twisting), and DNA extraction was carried out using a Qiagen MagAttract HMW kit (PN-67653) following the manufacturer’s instructions, with the following modifications: 200 ul 1X PBS was used in lieu of Buffer ATL; PBS was mixed simultaneously with RNAse A, Proteinase K, and Buffer AL prior to tissue homogenisation and incubation; incubation time was shortened to 2 h; solutions were mixed by gently flicking the tube rather than pipetting; and subsequent wash steps were performed for one minute. Any time DNA was transferred, wide-bore tips were used. These modifications were in accordance with recommendations from 10X Genomics HMW protocols that aim to achieve >50 kb molecules. The resulting sample contained ~250 ng of DNA, and we used the FEMTO Pulse (Advanced Analytical, Ankeny, IA, USA) to examine the molecular weight of the resulting DNA. This revealed a relatively sharp band at ~150 kb (Appendix A). The DNA was shipped from the U.K. to California on cold packs, and examined again by running 500 pg on the FEMTO Pulse. While a shift in the molecular weight profile was observed as a result of transport, showing a broader DNA smear with mode of ~40 kb (Figure 1), it was still suitable for library preparation (note that this shifted profile is coincidentally similar to what is observed with the unmodified MagAttract protocol). DNA concentration was determined with a Qubit fluorometer and Qubit dsDNA HS assay kit (Thermo Fisher Scientific, Waltham, MA, USA), and 100 ng from the 250 ng total was used for library preparation.

### 2.2. Library Preparation and Sequencing

A SMRTbell library was constructed using an early access version of SMRTbell Express Prep kit v2.0 (Pacific Biosciences, Menlo Park, CA, USA). Because the genomic DNA was already fragmented with the majority of DNA fragments above 20 kb, shearing was not necessary. 100 ng of the genomic DNA was carried into the first enzymatic reaction to remove single-stranded overhangs followed by treatment with repair enzymes to repair any damage that may be present on the DNA backbone. After DNA damage repair, ends of the double stranded fragments were polished and subsequently tailed with an A-overhang. Ligation with T-overhang SMRTbell adapters was performed at 20 °C for 60 min. Following ligation, the SMRTbell library was purified with two AMPure PB bead clean up steps (PacBio, Menlo Park, CA), first with 0.45X followed by 0.80X AMPure. The size and concentration of the final library (Figure 1) were assessed using the FEMTO Pulse and the Qubit Fluorometer and Qubit dsDNA HS reagents Assay kit (Thermo Fisher Scientific, Waltham, MA, USA), respectively.

Sequencing primer v4 and Sequel DNA Polymerase 3.0 were annealed and bound, respectively, to the SMRTbell library. The library was loaded at an on-plate concentration of 5–6 pM using diffusion loading. SMRT sequencing was performed on the Sequel System with Sequel Sequencing Kit 3.0, 1200 min movies with 120 min pre-extension and Software v6.0 (PacBio). A total of 3 SMRT Cells were run.

### 2.3. Assembly

The genome was assembled using FALCON-Unzip, a diploid assembler that captures haplotype variation in the sample ([7], see Appendix A for software versions and configuration details). A single subread per zero-mode waveguide (ZMW) was used for a total of 12.8 Gb of sequence from three SMRT Cells, or ~48-fold coverage of the ~266 Mb genome. Subreads longer than 4559 bp were designated as “seed reads” and used as template sequences for preassembly/error correction. A total of 8.1 Gb of preassembled reads was generated (~30-fold coverage). After assembly and haplotype separation by FALCON-Unzip, two rounds of polishing were performed to increase the consensus sequence quality of the assembly, aligning the PacBio data to the contigs and computing consensus using the Arrow consensus caller [8]. The first round of polishing was part of the FALCON-Unzip workflow and used a single read per ZMW that was assigned to a haplotype. The second round of polishing was performed in SMRT Link v 6.0.0.43878, concatenating primary contigs and haplotigs into a single reference and aligning all subreads longer than 1000 bp (including multiple subreads from a single sequence read, mean coverage 184-fold) before performing genomic consensus calling. The alignments (BAM files) produced during the two rounds of polishing were used to assess confidence in the contig assembly in regions with rearrangements relative to the AgamP4 PEST assembly for *Anopheles gambiae* (GenBank assembly accession GCA_000005575.2) [8,9]. We referred to the first round of polishing as using “unique subreads” and the second round as using “all subreads”.

We explored the performance as a function of the number of SMRT Cells used for the assembly (Appendix A), and found that while a single SMRT Cell was insufficient to result in high-quality assembly, data from two SMRT Cells generated a highly contiguous assembly of the correct genome size. We proceeded with the 3-Cell assembly for all subsequent analyses because it gave the best assembly results.

### 2.4. Curation

The contigs were screened by the Sanger Institute and NCBI to identify contaminants and mitochondrial sequence. Windowmasker was used to mask repeats and the MegaBLAST algorithm was run (with parameter settings: -task megablast -word_size 28 -best_hit_overhang 0.1 -best_hit_score_edge 0.1 -dust yes -evalue 0.0001 -min_raw_gapped_score 100 -penalty −5 -perc_identity 98.0 -soft_masking true -outfmt 7) on the masked genome versus all complete bacterial genomes to find hits with ≥98% homology. In addition, we screened the primary assembly for duplicate haplotypes using Purge Haplotigs [10] with default parameters and coverage thresholds of 20, 150, and 700.

In the process of using PEST to order and orient the PacBio contigs, we found one large potential heterozygous interchromosomal rearrangement between 2L and 3R (Appendix A). Upon further exploration, this was not supported by any subreads mapping across the breakpoint (Appendix A). The putative breakpoints were identified by aligning the PacBio contigs to PEST with minimap2 (asm5 setting), and the start and end position of each aligned subread was determined using bedtools ‘bamtobed’. This 4.9 Mb contig had no reads spanning the putative breakpoint when either “unique” or “all subread” alignments were examined and thus we designated this a chimeric misassembly, and split the contig into two.

### 2.5. Genome Quality Assessment

To assess the completeness of the curated assembly, we searched for conserved, single copy genes using BUSCO (Benchmarking Universal Single-Copy Orthologs) v3.0.2 [11] with the dipteran gene set. In addition, we evaluated assembly completeness against a curated set of genes (AgamP4.10 gene set) from the *An. gambiae* PEST assembly, using a previously described script [12].

To assess the quality of contig assembly and concordance with existing assemblies, the curated primary contigs were aligned to the PEST *Anopheles gambiae* reference genome [8,9] using minimap2 with the “map-pb” settings [13]. For the purpose of comparison, contigs were ordered and oriented according to their median alignment position and orientation on their majority chromosome. A python script with pysam was used in conjunction with ggplot using geom_segments to generate the alignment plots. Large regions (≥250 kb) where assembly contigs did not align to PEST, or where multiple contigs aligned to the same reference region, or where large portions of a single contig aligned discordantly (e.g., to multiple reference chromosomes) were identified and explored manually by visualizing questionable alignments and their breakpoints in the Integrated Genome Browser (IGV, [14]). Confidence in contig assembly was assessed by evaluating subread mapping across putative rearrangement breakpoints as described above. For subread coverage plots, alignments were also made using minimap2 with the “map-pb” setting, and a smoothing filter was applied (mapq 60 filter averaged in 5 kb bins for Figure 3 and Appendix A, and mapq 60 filter averaged in 50 kb bins for Figure 4, respectively) using a custom python script and pysam/numpy. All python scripts referred to above are available [15].

## 3. Results

### 3.1. A Modified Protocol Allows for Library Preparation and Sequencing of Samples from as Low as 100 ng of DNA Input

High molecular weight DNA was extracted from a single female mosquito. Given that the genomic DNA had a suitable size range for long-insert PacBio sequencing (Figure 1), the sequencing library preparation protocol was modified to exclude an initial shearing step, which facilitated the use of lower input amounts, as shearing and clean up steps typically lead to loss of DNA material. After following the Express template preparation protocol, the final clean up step was simplified to just two AMPure purification steps to remove unligated adapters and very short DNA fragments, resulting in a final library with a size distribution peak around 15 kb (Figure 1). The library was then sequenced on the Sequel System on 3 SMRT Cells, generating on average 24 Gb of data per SMRT Cell, with average insert lengths of 8.1 kb (insert length N50 ~13 kb, Appendix A). The overall library yield was 59%, which would have allowed for the sequencing of at least 8 SMRT Cells, thereby potentially allowing for genome sizes 2–3 times larger than studied here in conjunction with this protocol.

### 3.2. De novo Assembly Using FALCON-Unzip Allows for a High-Quality Genome from a Single Anopheles coluzzii Mosquito Individual

Using the FALCON-Unzip assembler [7], the resulting primary *de novo* assembly consisted of 372 contigs totaling 266 Mb in length, with half of the assembly in contigs (contig N50) of 3.5 Mb or longer (Table 1). FALCON-Unzip also generated 665 alternate haplotigs, representing regions of sufficient heterozygosity to allow for the separation of the maternal and paternal haplotypes. These additional phased haplotype sequences spanned a total of 78.5 Mb (i.e., 29% of the total genome size was separated into haplotypes), with a contig N50 of 223 kb (Table 1). One contig (#20) was identified as a complete 4.24 Mb bacterial genome, closely related to *Elizabethkingia anophelis*, which is a common gut microbe in *Anopheles* mosquitoes [16]. It was separated from the mosquito assembly and submitted to NCBI separately (see availability of data). We also identified two contigs of mitochondrial origin that each contained multiple copies of the circular chromosome. Full length copies of the mitochondrial chromosome in the higher quality contig differed by only a single base and the consensus sequence was reported as the mitochondrial genome. One of these copies was discarded.

While FALCON-Unzip resolved haplotypes over ~30% of the genome, 110 genes appeared as duplicated copies in the BUSCO analysis, indicating that highly divergent haplotypes may be assembled as distinct primary contigs as has been observed in other mosquito genome assemblies [6,18]. The presence of duplicated haplotypes can result in erroneously low mapping qualities in resequencing studies and cause problems in downstream scaffolding. Using the “Purge Haplotigs” software [19], we identified 165 primary contigs totalling 10.6 Mb as likely alternate haplotypes, although there remains a possibility that some may be repeats. These contigs were transferred to the alternate haplotig set.

After the above curation steps, including the removal of the bacterial contig, the haplotigs, and the extra copy of the mitochondrial genome, as well as splitting the large chimeric contig, the primary assembly consisted of 206 contigs totaling 251 Mb with contig N50 of 3.47 Mb. Compared to the Sanger sequence based assembly for *An. coluzzii* [17] (AcolM1 Mali-NIH strain assembly AcolM1; GenBank assembly accession GCA_000150765.1), this translated to a reduction in the number of contigs by ~130-fold, as well as an increase in genome contiguity by ~140-fold (Table 1). The PacBio primary assembly was also 12% larger in total size, reflecting additional genomic content that was missing in the previous assembly, corroborated by the conserved gene analysis (see BUSCO analysis results below).

To evaluate genome completeness and sequence accuracy of the assembly, we performed alignment analyses to a set of conserved genes. Using the ‘diptera’ set of the BUSCO gene collection [11], we observed 98% of the ~2800 genes were complete and >95% occurred as single copies (Appendix A). By comparison, the previous assembly had 87.5% complete BUSCO alignments, indicating that a fraction of the genome was missing in that assembly. The percentage of duplicated genes was reduced from 3.9% to 2.4% after curation. Additional analyses are required to distinguish true gene duplication events from incomplete purging of duplicated haplotypes (see discussion below and Appendix A). As an additional evaluation, we aligned to the primary assembly a closely related species gene set (the most recent *An. gambiae* (AgamP4.10) gene set), resulting in 14,972 alignments (99.5%) and an average alignment length of 96.6%, and with >96% of alignments showing no frame shift-inducing indels.

### 3.3. The New Assembly Shows Improvements in Resolving Genomic Regions

The *An. gambiae* genome, published in 2002, was created using BACs and Sanger sequencing [8]. Further work over the years to order and orient contigs improved this reference [9,20] and to date, AgamP4 [21] remains the highest quality *Anopheles* genome among the 21 that have now been sequenced [4]. However, AgamP4 still has 6302 gaps of Ns in the primary chromosome scaffolds ranging from 20 bases to 36 kb, including 55 gaps of 10 kb that the AGP (A Golden Path) file on Vectorbase annotates as ‘contig’ endings. The AgamP4 genome was generated from a lab strain known as PEST (Pink Eye STandard) that is long deceased and also was an accidental mixture of two incipient species, previously known as “M” and “S”. To address this, the genomes of pure “M” and “S” from new colonies established in Mali were sequenced using only Sanger sequencing [17]. Since then, the “S” form has retained the name *An. gambiae sensu stricto*, and the “M” form has acquired species status and a new name, *An. coluzzii* [22]. It is important to note that while these species show assortative mating, they can hybridise in nature and their hybrids are fully fertile and viable [23]. Given this fact, and the fact that both pure species assemblies remain highly fragmented, we compared our assembly to the best available *An. gambiae* genome (i.e., AgamP4 PEST [21]) to evaluate contiguity and to help order and orient the contigs.

The new PacBio assembly is highly concordant with the AgamP4 PEST reference over the entire genome, allowing the placement of the long PacBio contigs into chromosomal contexts (Figure 2). In addition, the high contiguity of the PacBio contigs allows for the resolution of many gaps in the chromosomal PEST ‘contigs’. Note that the only gaps in the PacBio assembly are at contig ends, whereas there are many gaps in PEST that are not annotated as contig breaks so the percent Ns per megabase of PEST is overlaid in the graphs in Figure 2. For example, a single contig from the new PacBio assembly expanded a tandem repeat region on chromosome 2L that in PEST was collapsed, while also filling in many Ns (gaps) in PEST, and also spanning a break between PEST scaffolds set to 10,000 Ns (Figure 3).

The PEST annotation also retains a large bin of unplaced contigs (27.3 Mb excluding Ns) designated as the “UNKN” (unknown) chromosome. We compared the alignments of contigs from the PEST chromosomes (X, 2, 3) versus the contigs from the UNKN to the new assembly. Any regions with mapq60 alignments of both UNKN and chromosomal contigs are likely to be haplotigs in the UNKN. In total, we find that 7.27 Mb are haplotigs (i.e., also have PEST chromosomal alignments to the same location in the assembly) and another 10.9 Mb are newly placed sequence that do not overlap with PEST chromosomal alignments. The UNKN bin also contains 737 annotated genes. Remarkably, our single-insect assembly now places 667 (>90%) of these formerly unplaced genes into their appropriate chromosomal contexts (2L:148 genes; 2R:162 genes; 3L: 126 genes; 3R:91 genes; X:140 genes; unplaced:70 genes; details on specific genes can be found in Appendix A), which together with their flanking sequence comprise 8.9 Mb of sequence. Altogether, this means that 40% of the UNKN chromosome is now placed in the genome, along with 90% of the genes that were contained within it.

We also identified several potential rearrangements in the 20–22 Mb region of the X chromosome (Figure 4). PEST has contig breaks at the putative breakpoints relative to the assembly, however, given that a single PacBio contig spans the full region and that potential breakpoints relative to PEST are supported by multiple reads, the most likely explanation is an order and orientation issue in PEST, perhaps combined with a potential inversion difference between *An. coluzzii* and the PEST reference. In addition, the contig contains a relatively large region (~380 kb in total) of PacBio sequence corresponding to several pieces in the UNKN section of PEST that can now be assigned to the X chromosome.

## 4. Discussion

Long-read PacBio sequencing has been utilized extensively to generate high-quality eukaryote *de novo* genome assemblies, but because of the relatively large DNA input requirements, it has not been used to its full potential for small organisms, requiring time-consuming inbreeding or pooling strategies to generate enough DNA for library preparation and sequencing. Here we present, to our knowledge, the first example of a high-quality *de novo* assembly from a single insect. This assembly, using only one individual and one sequencing technology, exhibits a higher level of contiguity, completeness, accuracy, and degree of haplotype separation than any previous *Anopheles* assembly, demonstrating the impact of long reads on assembly statistics. While the assembly did not achieve independent full chromosomal scale assignment of contigs, its mega-base scale contiguity without gaps immediately provides insights into gene structure and larger-scale genomic architecture, such as promoters, enhancers, repeat elements, large-scale structural variation relative to other species, resolution of tandem repeats (Figure 3), and many other aspects relative to functional and comparative genomics questions.

About a third of the genome for this diploid individual is haplotype-resolved and represented as two separate sequences for the two alleles, thereby providing additional information about the extent and structure of heterozygosity that was not available in previous assemblies, which have been constructed from many pooled individuals. In contrast with approaches requiring multiple individuals, the ability to generate high-quality genomes from single individuals greatly simplifies the assembly process and interpretation, and will allow far clearer lineage and evolutionary conclusions from the sequencing of members of different populations and species. Further, if parental samples are available, the recently developed trio binning assembly approach [24] can be used to further segregate alleles for a full haplotype-resolved assembly of both parental copies of the diploid offspring organism.

The assembly presented here provides an excellent foundation towards generating an improved chromosome-scale reference genome, using the previous PEST reference, scaffolding information from genetic maps, technologies such as Hi-C (e.g., [25]), or alignment of the contigs to closely related species’ references. These approaches can also be used to highlight areas of potential improvements to the FALCON-Unzip assembler and to Purge Haplotigs, or other packages used to identify haplotypic contigs. As one example, we noticed in the context of the incomplete haplotype purging described above that some neighboring contig ends exhibited overlaps relative to the PEST reference (Appendix A). The interpretation of such haplotype contig overlaps was corroborated by the observed halving of average sequencing depth over the regions of overlap. These methods could incorporate adjustments to try to account for haplotypic regions in the ends of contigs rather than complete contigs being fully haplotypic.

We noted the importance of the initial DNA size distribution in conjunction with this protocol. Since neither shearing prior to library construction nor size-selection thereafter were employed, the starting high-molecular weight DNA should contain fragments at greater than ~20 kb on average, and without the significant presence of short (smaller than ~5 kb) DNA fragments. Further research into suitable DNA extraction, storage and transportation methodologies is needed to fulfill these requirements for a broader spectrum of different species and environments, in order to allow for the preparation of suitable DNA samples from wild-caught samples originating in sometimes remote areas with limited sample preparation infrastructure.

We anticipate that the new workflow described here will facilitate the sequencing and high-quality assembly of many more species of small organisms, as well as groups of individuals within a species for population-scale analyses, representing an important prerequisite in view of large-scale initiatives such as i5K and the Earth BioGenome Project [1,5]. In addition, other research areas with typically low DNA input regimes could benefit from the described new workflow, e.g., metagenomic community characterizations of small biofilms, DNA isolated from needle biopsy samples, minimization of amplification cycles for targeted or single-cell sequencing applications, and others.

## Figures and Tables

**Figure 1 genes-10-00062-f001:**
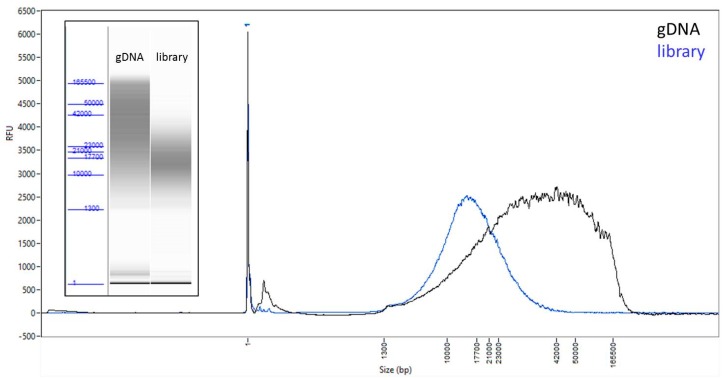
*Anopheles coluzzii* input DNA and resulting library. FEMTO Pulse traces and ‘gel’ images (inset) of the genomic DNA input (black) and the final library (blue) before sequencing.

**Figure 2 genes-10-00062-f002:**
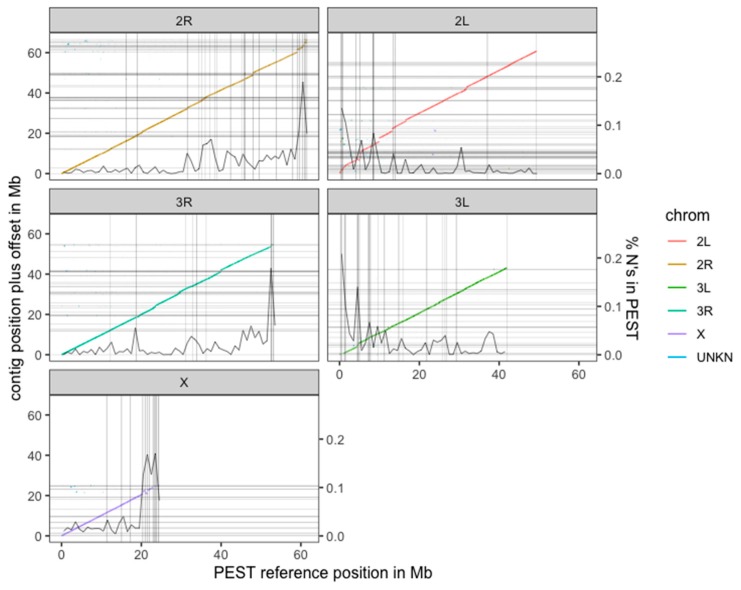
Alignment of the curated PacBio contigs to the AgamP4 PEST reference [21]. Alignments are colored by the primary PEST reference chromosome to which they align but are placed in the panel and *Y* offset to which the contig as a whole aligns best. Contig ends are denoted by horizontal lines in the assembly and vertical lines in PEST. However, there are many Ns in PEST not annotated as contig breaks so the percent Ns per megabase of PEST is overlaid (scale on the right *Y* axis). There are no Ns in the PacBio assembly.

**Figure 3 genes-10-00062-f003:**
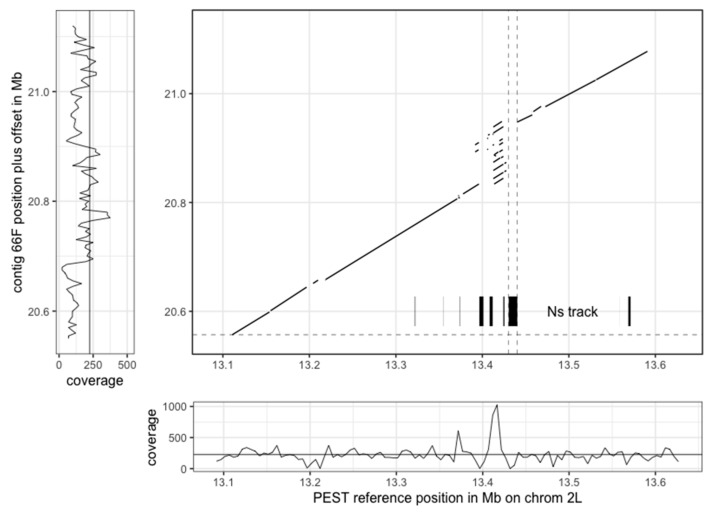
Example of a compressed repeat in PEST that has been expanded by the PacBio assembly. Dotted vertical lines represent a gap in the PEST assembly (10,000 Ns) between scaffolds, which is now spanned by the single PacBio contig. Coverage plot of the PacBio subreads aligned to PEST (bottom) highlights the region where excess coverage indicates a collapsed repeat in PEST, in contrast the coverage of PacBio subreads aligned to the PacBio contig (left) is more uniform.

**Figure 4 genes-10-00062-f004:**
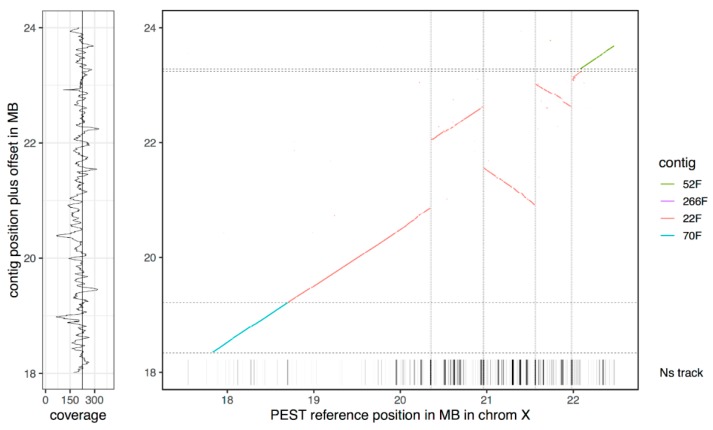
Alignment of X pericentromeric contigs to PEST, highlighting likely order and orientation issues in the PEST assembly that are resolved by a single PacBio contig (22F).

**Table 1 genes-10-00062-t001:** Assembly statistics of raw and curated PacBio *Anopheles coluzzii de novo* assembly, compared with the previous Sanger-sequence based assembly for this species from [17] (GCA_000150765.1).

		PacBio Raw	PacBio Curated	Sanger Assembly
**Primary contig assembly**	**Size (Mb)**	266	251	224
	**No. contigs**	372	206	27,063
	**Contig N50 (Mb)**	3.52	3.47	0.025
**Alternate haplotigs**	**Size (Mb)**	78.5	89.2	unresolved
	**No. contigs**	665	830	N/A
	**Contig N50 (Mb)**	0.22	0.199	N/A

## Data Availability

Raw data and assemblies are deposited in NCBI under BioProject PRJNA508774, and at https://downloads.pacbcloud.com/public/dataset/Mosquito_singleFemale_Assembly/.

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
