# Peer review of "A High-Quality De novo Genome Assembly from a Single Mosquito Using PacBio Sequencing"

_genes, 2019, doi:10.3390/genes10010062_

Round 1
Reviewer 1 Report
The article of S. Kingan and co-authors describes a high-quality genome assembly developed for a malaria vector, Anopheles coluzzii. The new assembly was constructed from a single female mosquito using long read the Pacific Bioscience (PacBio) sequencing technology. This assembly represents a significant leap forward compared with the previously developed genome assemblies for An. gambiae and An. coluzzii that utilized Sanger and Illumina sequencing technologies. The methods of DNA extraction, sequencing and assembly construction are described in details. In addition, the new genome of An. coluzzii is compared to the An. gambiae PEST genome assembly. Importantly, the authors were able to resolve maternal and paternal haplotypes for over 1/3 of the genome. The study demonstrates that PacBio sequencing is a promising technology for the development high-quality genome assemblies for small organisms such as insects.
I have minor suggestions for improving the manuscript.
1) It would be helpful to add an additional figure to the discussion section to visualize the overall assembly improvement over the PEST genome assembly. It is important to indicate the regions on the chromosomes where the UNKN scaffolds from the PEST genome assembly belong. In the current description, it is unclear if they localize in telomeres, centromeres or in the middle of the chromosome arms.
2) Telomeres and centromeres were not assembled in the previous Anopheles genomes. It is important to provide information on these chromosome regions in the new An. coluzzii genome assembly if they were captured.
3) There is no mentioning of providing the new assembly to VectorBase that could make it easily accessed by other researches.
4) There is a dot before the word Discussion (lane 298) that has to be removed.
Author Response
We would like to thank the reviewers for their positive interest in our work and for their very helpful suggestions, which we have addressed in full in the revised manuscript. Please see the detailed response to each suggestion in the attached file, we have highlighted the corresponding changes in the revised manuscript appropriately.

Reviewer 2 Report
The development of new protocols to sequence long reads (PacBio) from significantly less material is an exciting leap forward for generating genome assemblies from single organisms, the application of this method to Anopheles mosquitoes is both timely in terms of prospective new arthropod sequencing and helpful in terms of the, now quite old, PEST assembly for Anopheles gambiae. The paper clearly sets out the authors goals with plenty of detail about assembly parameters etc. which presumably would allow interested parties with sufficient bioinformatics knowledge to run the assembly algorithm, that is the novelty here is the library preparation and not the downstream informatics.
The use of the An. gambiae PEST assembly as a comparator for assessing the accuracy of the PacBio Anopheles coluzzii assembly is appropriate but at times the focus shifts between validating the An. coluzzii assembly (e.g. BUSCO statistics) and documenting issues and remedial suggestions for the An. gambiae PEST assembly. It would be good to see more context as to why these issues have been highlighted, e.g. advantages of long reads in resolving (collapsed) tandem repeats or potential problems of chimeric contigs (Figure S2).
Other comments:
Consider moving some of the details about existing An. gambiae and An. coluzzii genome assemblies from Section 3.3 as this provides the context/framework within the results are to be assessed e.g. comparison of assembly statistics in Table 1.
There are several references to custom python scripts and graphing libraries used in the generation of the figures. Given that the code is not available, that I can see, I don't think this information adds to the paper and suggest either that these references are removed or the code made available.
Section 2.5 mentions performing CEGMA analysis as a metric for protein-coding gene completeness. As the authors are aware this has been replaced by BUSCO and there is no need to do both. The fact that the CEGMA results are not included in the paper or supplemental data would appear to support this view.
Were all the 165 contigs identified as alternative haplotigs and removed from the primary assembly identified using the 'Purge Haplotypes' software or was the manual assessment of alignments noted in Section 2.5 implicate any contigs for removal? In some ways 'Curated' implies manual effort and not a secondary, automated filtering step.
There is a mismatch in the number of contigs between the 'Raw' and 'Curated' assemblies in Table 1. Presumably this is due to the removal of the Elizabethkingia anophelis bacterial contig but it is not explicitly stated and it is not clear whether the authors left multiple copies of the mtDNA in the assembly.
Consider other metrics of contiguity for the assembly such as NG50 to normalise across assembly versions.
Section 3.3 has a number of mentions of 'Anopheles gambiae', convention is to write the species name in full for the first mention and then abbreviate for subsequent mentions.
If possible it would be good to use consistent nomenclature for the genome assemblies. I note that the An. gambiae PEST assembly is referred to by the VectorBase identifier AgamP4 but the An. coluzzii Mali-NIH assembly is referred to by the INSDC accession GCA_000150765. Perhaps include both.
The paragraph on assignment of chromosomal location of UNKN contigs from the PEST assembly (lines 275-282) is not directly relevant to the paper in that it reflects problems in the An. gambiae PEST assembly and not perceived improvements of the PacBio An. coluzzii assembly. As an example many of the unplaced contigs are alternative haplotigs that were not removed from the PEST assembly, now 16 years old. A more interesting metric here would be how many of the contigs placed onto the An. coluzzii assembly are unique as opposed to potential haplotigs.
Assembly completeness as measured via BUSCO statistics is useful when sequencing new species for which no existing gene set is available. This is not the case for this study where there is a curated set of genes for the An. gambiae PEST assembly that should be highly similar. Metrics based on the gambiae gene predictions could inform about potential indel errors in the assembly, a potential issue for PacBio assemblies.
Checking INSDC, the assembly for Elizabethkingia is available but not An. coluzzii. My feeling is that there is no need to maintain/harmonise contig names in the paper with INSDC accessions but the authors may want to include Assembly accessions as they become available to supplement the BioProject accession.
Author Response

(The authors gave the same response as above.)
